# Controlling the Heterodimerisation of the Phytosulfokine Receptor 1 (PSKR1) via Island Loop Modulation

**DOI:** 10.3390/ijms22041806

**Published:** 2021-02-11

**Authors:** João V. de Souza, Matthew Kondal, Piotr Zaborniak, Ryland Cairns, Agnieszka K. Bronowska

**Affiliations:** 1School of Natural and Environmental Sciences, Newcastle University, Newcastle upon Tyne NE1 7RU, UK; Matthew.kondal@newcastle.ac.uk (M.K.); p.zaborniak@newcastle.ac.uk (P.Z.); 2Fontus Environmental, High Garth, Thirsk YO7 3PX, UK; ryland@fontusenvironmental.com

**Keywords:** phytosulfokine receptor 1, island loop, metadynamics, molecular dynamics, crop control, Plant Hormone

## Abstract

Phytosulfokine (PSK) is a phytohormone responsible for cell-to-cell communication in plants, playing a pivotal role in plant development and growth. The binding of PSK to its cognate receptor, PSKR1, is modulated by the formation of a binding site located between a leucine-rich repeat (LRR) domain of PSKR1 and the loop located in the receptor’s island domain (ID). The atomic resolution structure of the extracellular PSKR1 bound to PSK has been reported, however, the intrinsic dynamics of PSK binding and the architecture of the PSKR1 binding site remain to be understood. In this work, we used atomistic molecular dynamics (MD) simulations and free energy calculations to elucidate how the PSKR1 island domain (ID) loop forms and binds PSK. Moreover, we report a novel “druggable” binding site which could be exploited for the targeted modulation of the PSKR1-PSK binding by small molecules. We expect that our results will open new ways to modulate the PSK signalling cascade via small molecules, which can result in new crop control and agricultural applications.

## 1. Introduction

Molecular signalling exerted through peptide hormones plays a key part in regulating plant growth and reproduction [1,2,3]. These phytohormones are also responsible for an environmental response crucial to plant survival and adaptation to different conditions [2,4]. One of such hormones is phytosulfokine (PSK): A sulphated peptide which is biologically active in nanomolar concentrations [5,6]. Comprised of five residues (Tyr(SO_3_H)-Ile-Tyr(SO_3_H)-Thr-Gln) [7,8], of which both tyrosines are sulphated, PSK has been classified as a crucial plant growth factor [3,9,10].

PSK is matured by the proteolytic cleavage of its precursors and then goes through a post-translational sulfanation of its tyrosine residues [11]. Its main binding partners are the phytosulfokine receptors (PSKR) [12,13]. These receptors were first identified on *Daucus carrota*, and its gene is conserved among different plants species [7,10]. One of these is *Arabidopsis thaliana,* which expresses two different orthologues: PSKR1 [14] and PSKR2 [15], with the PSK sensing occurring mainly through PSKR1 [15].

PSKR1 belongs to the leucine-rich repeat receptor kinases (LRR-RK) family. It is comprised of two large domains: Residues 75 to 619 contain the leucine-rich repeat (LRR) domain, and the region located within residues 734 to 1005 comprises the protein kinase domain. In 2015, Wang and co-workers solved the crystal structure of a series of different PSKR1 LRR domain configurations [16]. They showed that the PSKR1 LRR domain is comprised of a large right-handed superhelical region, which is comprised of 20 LRRs, and an island domain (ID) [16]. They also solved the structure of the PSKR1 LRR domain bound to PSK and the PSKR1/LRR-PSK complex bound to the somatic embryogenesis receptor-like kinase 1 (SERK1).

The PSKR1 LRR domain binds its partner through a different mode than other hormone-dependent LRR-RK. For example, the brassinosteroid-insensitive 1 (BRI1) [17,18] and the flagellin insensitive 2 (FLS2) [19] are LRR-RKs which are also comprised of a right-hand LRR superhelix. These three proteins are structurally similar, and their signalling pathways are highly related [20]. Both BRI1 and FLS2 bind their respective hormones (brassinolide or flg22) as heterodimers, with the binding site occurring between LRR-RK and their respective heterodimerisation partner. They are receptor kinases for BRI1-associated kinase 1 (BAK1) and have a role in mediating the LRR–BAK1 interaction [18,21]. The signalling events mediated by these two receptors require their ligands to be bound within the dimerization interface. In the event of a PSK-PSKR1-SERK1 complex forming, the PSK interacts solely with PSKR1. The binding event stabilises the loop within the island domain on PSKR1, without any direct interactions between PSK and SERK1 [16], but the details of the formation of this complex remain poorly understood.

In this work, we used atomistic molecular dynamics (MD) simulations and enhanced sampling techniques to study how the PSKR1-PSK binding event occurs at the nanoseconds to microseconds time scale. We have proposed a sequential model of PSKR1-PSK binding, which goes through a transient, local minimum denoted as state B. We have shown that state B creates a novel allosteric “druggable” binding site, the formation of which is dependent on PSK. This binding site shows characteristics which can be used for the structure-guided design of small molecular modulators of PSK-PSKR1 signalling, which may prove useful for agricultural applications [16].

## 2. Results

To investigate the events behind the PSK-PSKR-SERK1 complex formation, equilibrium simulations were made based on the available crystal structures. Simulations for the apo DcPSKR1 were performed, with the island loop (IL) of the island domain in an unstructured conformation. As shown in Figure 1, the root-mean-square deviation (RMSD) of the DcPSKR1-apo displays higher fluctuations in comparison to the PSKR1-holo and the PSKR1-apo (Figure 1A). This difference in intrinsic flexibility arises mainly from the island loop comprised of residues 501 to 536 (Figure 1B,C), which regulates the binding of the PSKR1. The apoPSKR1 did not form a β-sheet, which was found in the holo-structure. Instead, the apo ID loop transitioned from a partially helical/disordered conformation to a fully disordered configuration, which resulted in a higher radius of gyration for the apo ID loop than its holo counterpart (Appendix A).

Analysis of the total number of hydrogen bonds (H-bonds) shows a decrease through the simulation time, reaching a peak of six H-bonds formed (Figure 2), indicating that the DcPSKR1-apo configuration of the island domain does not interact tightly with the LLR domain. The average number of H-bonds between LRR and island domain was three, resulting in a higher solvent accessible surface area (SASA) for the 501–536 loop (Appendix A). This increase in the SASA is due to the high net charges located within this loop, such as the poly-lysine region (residues 517–519), and a high prevalence of disorder inducing residues like the G524 and G525. In comparison, the PSKR1-PSK complex has a highly constrained loop, with a lower RMSF than its apo counterparts (Appendix A). The β-sheet located in the island loop interacts with the PSK backbone, stabilising the interaction. The PSK backbone acts as a third β-strand configuration and forming a series of hydrogen bonds between the F506-R509 and the peptides that comprise the PSK.

The sulphated tyrosines are fundamental for binding PSK to PSKR1. The two sulphated tyrosines of PSK create a hydrogen bond network conserved for both residues, as shown in Figure 2A,B. The N-terminal PSK tyrosine interacted with K509 and N424 with an average heavy atom distance of 0.35 ± 0.02 nm and 0.38 ± 0.02 nm, respectively, for the first 200 ns of the simulation. Interestingly, we observed a swap of interactions: The sulphate group flipped between interactions with N424 and K508 (Figure 3A), and while the other sulphated tyrosine also interacted with K508 (D_S-NZ_ = 0.36 ± 0.02 nm in the first half of the simulation), it also found another stable interaction with R349. The starting distance between the S atom of sulphated tyrosine and the NH2/NH1 of R349 was 0.6 nm, but it has been reduced to 0.36 ± 0.02 nm during the simulation, as shown in Figure 3B. This chain (N424- > S-Tyr- > K506- > S-Tyr- > R349) connected residues form the island loop to the LRR-domain via the PSK sulphates, resulting in a low RMSD value in comparison to the crystallographic structure (0.23 ± 0.04 nm for the loop region and 0.2 ± 0.03 for the whole protein). This also directly affected the PSK crystallographic pose, which has an average of 0.05 ± 0.01 nm for its RMSD.

The simulations of *apo*-PSKR1 starting from the folded configuration showed an unstable folded loop. The equilibrium run for the structured apo PSKR1 showed that after 200 ns, the island loop started to acquire a different state from its initial configuration and given by the fact that the RMSD before 200 ns has an average of 0.27 ± 0.05 nm and then between 200 ns to 500 ns, the value increased to 0.55 ± 0.05 nm. Nonetheless, no equilibrium replica showed a complete unfolding of the island loop in the simulation of apo-PSKR1 (Appendix A).

### 2.1. Metadynamics of Island Loop Transition

To evaluate the loop transition towards different local free energy minima, well-tempered metadynamics simulations were carried out. Starting from the holo configuration and using the radius of gyration of the island loop (RG_ID_) as the collective variable, the second stable conformation was found. The metadynamics free energy surface is shown in Figure 4. A second folded state (state B, RG_ID_ = 1.75 nm) was found as a local minimum, with a difference of 20 kcal/mol in comparison to the native state (RG_ID_ = 1.3). These 2 states are shown in Figure 5B. This energy difference arises from the partial breaking of the interactions between the loop and the PSK. Transitioning from state B, the energy barrier to folding back to the native conformation is 3 kcal/mol. The PSK-PSKR1 interactions in state B occur in three different PSK regions: The two PSK termini and the N-terminal sulphated tyrosine. While the N terminal interaction is kept between the terminus backbone and D445, the PSK C-terminus keeps its interaction with R300. The N-terminal sulphated tyrosine keeps its interaction with the N424. Nonetheless, the network of hydrogen bonds was broken, reducing the number of hydrogen bonds compared to the equilibrium run, thus disrupting the PSK-ID loop stability, as shown in Figure 5A.

While in state B, the number of interactions between PSKR1 and PSK is reduced. PSK does not fully unbind in state B: However, the average number of hydrogen bonds decreased from six in the equilibrium runs to four in state B.

The formation of state B is PSK-dependant. To better understand the formation of state B, we ran well-tempered metadynamics from the holo configuration without the presence of the PSK. As shown in Figure 4, the energy well located at RG_ID_ = 1.75 nm did not appear as a local minimum. Throughout the metadynamics, the ID loop steadily unfolded towards an unstructured conformation (Appendix A), resulting in a disordered loop after 60 ns of metadynamics. This indicates that the formation of state B depends on the partial stabilisation caused by the residual interactions of PSKR1-PSK.

The island loop conformation for state B is different from the native PSK-PSK1-SERK1 configuration. As shown in Figure 5B, the PSKR1-PSK experimental structure complex is different from the obtained configuration in state B. The translation of the loop towards the area where SERK1 binds to PSKR1 creates a series of steric clashes (Figure 6), impeding the biological function of PSKR1. Hence, the allosteric binding site observed exclusively in state B may result in an antagonist modulation of the SERK1 binding, disrupting the PSKR1 signalling cascade. The convergence of the systems for the well-tempered metadynamics is shown in Appendix A.

### 2.2. Evaluation of State B “Druggability”

To evaluate the possible novel binding sites observed exclusively in state B, solvent mapping by FTMAP (https://ftmap.bu.edu/, Accessed in 14 October 2020) webserver was carried out on both configurations. The results of which, shown in Figure 7, identified a novel binding site that is not found in native PSKR1. The cavity identified by FTMAP is formed in the region opposite to the PSK, between the LRR domain and the island loop.

This novel binding site is comprised of residues S320, D321, and L296 on the LRR domain and residues D502, F503, R514, and Q517 (Figure 8) of the central region of the island loop. This binding site shows a series of characteristics: First, the loop side is composed of a sequence of polar residues. Second, it has a slightly hydrophobic patch, open for apolar interactions, specifically on the LRR domain side. The small molecular probes found by FTMAP to interact with this site were acetamide, acetaldehyde, benzaldehyde, phenol, cyclohexane, and benzene. Therefore, most of them can interact with both the apolar patch and the polar residues located in the island loop.

The existence of this binding site does not fully obstruct the main PSK binding site. This is evidenced by the fact that PSK did not fully unbind after the loop transition, strongly indicating that this is a non-competitive binding site.

To explore the possibility of this non-competitive allosteric binding site as a target for structure-based drug design, we docked the Enamime Essential Fragment library (320 small molecule fragments). This library was designed in collaboration between Enamine and the University of Cambridge, UK, to be a universal tool for the initial screen of novel targets. All compounds in the library are commercially available and have been tested for water solubility and chemical stability in buffer solution.

After the molecular docking, binding affinity calculation, and visual inspection of the poses, we selected four interesting fragments, suitable for fragment growth and lead optimization. These fragments are showed in Figure 9. They form hydrogen bonds with residue Q517 and hydrophobic interactions with the LRR-side of the binding site. All four fragments had mid µM predicted affinity despite their small molecular size. Moreover, these molecules showed that there is a deeper hydrophobic area, occupied by the dimethyl part of 4,4-dimethyl-1,2,3,4tetrahydroisoquinoline (Figure 9A) and the phenyl of the 3,5-dimethyl-4-phenyl-isoxazole (Figure 9D). We also found that indole-3-acetic acid (IAA) weakly bound (predicted mid-μM range) to that site (Appendix A). Similarly to other fragments shown in Figure 9, IAA formed H-bond with Q517 and aromatic interactions with W253. We followed this up, finding out that three other auxins: 2-phenylacetic acid (PAA), indole-3-propionic acid (IPA), and indole-3-butyric acid (IBA) also bound to this site with weak (predicted mid-μM range) potency.

## 3. Discussion

### PSK Directly Affects the Folding Energy of the Island Loop

The PSKR1 simulations of PSKR1 showed a significant difference in dynamics for the island loop. The apo unstructured loop transitioned between a conformation with a low helical content towards a fully unstructured/disordered loop. This indicates that this region is disorder-prone, which was expected given the fact that this region has several disorder-inducing motifs, such as the triple lysine repeat. Moreover, the disordered form of the ID loop transits randomly between configurations, resulting in a series of nonspecific interactions with the LRR repeats of PSKR1.

The decrease of secondary structure content coupled to high spatial fluctuation observed in all replicas indicates that PSK binding is required to induce the secondary structure of the ID loop and adopt a β-turn-β conformation. This is consistent with findings by Wang et al. [16] on the *Daucus Carota* PSKR (DcPSKR1) free receptor. The crystal structure of the free DcPSKR1 does not show density for the island loop, and our simulations show that area as fully flexible and highly unstable. The PSKR1-PSK allows for heterodimerisation differently than brassinosteroid-insensitive 1 (BRI1) and the flagellin insensitive 2 (FLS2). The model of activation of the heterodimer and their respective kinase is different: Both FLS2 and BRI1 mediated the LRR–BAK1 interaction via a series of residues located in the LRR domain and with interactions with their respective interaction modulators, flg22 and BLD. This is different from how the PSK modulates heterodimerisation of PSKR1, given the fact that the island loop changes its conformation in presence of PSK, controlling the interaction of SERK1 with PSKR-LRR. The interactions which mediate the PSKR1-SERK1 binding are located in the LRR domain, specifically with residues F596, S598, T619, and S623. However, the ID loop must maintain a high internal organisation for a stable SERK1 binding event. Therefore, understanding the dynamics under which the PSK-PSKR1-SERK1 complex formation occurs may aid in the development of innovative, small molecule-based agricultural applications for crop control.

The PSKR1 simulations with a folded ID loop and without the PSK showed that the island loop is highly structurally unstable, allowing the loop to start accessing disordered states.

The binding of PSK has a negligible effect on the overall LRR domain dynamics, but does affect the LRR-domain residues that interact with the ligand. LRR domains are known to be structurally stable, which correlates with the low RMSD value sampled in this area for both the apo and holo simulations. On the apo disordered loop, its elongated conformation randomly and transiently interacts with the LRR domain, but nothing that has a significant effect on the dynamics overall. On the other hand, PSK has several features that are crucial for loop stabilisation. The sulphated tyrosine and the PSK backbone atoms are key for PSK-PSKR1 interactions, suggesting that these interactions are enthalpy-driven by Coulombic/hydrogen bond interactions. These result in a highly specific and favourable contribution to binding free energy. Interestingly, this binding site is found exclusively within the PSKR1, without any requirement for the binding partner. This contrasts with the BRI1-BLD-BAK1/FLS1-flg22-BAK1 complex formation since the ligand is bound between two proteins forming a heterodimer. Therefore, understanding the PSKR1-PSK binding mechanisms should be independent of any tertiary partner. Hence, this emerged as an interesting target for function modulation via controlling the dynamics of the loop, and subsequently, controlling the signalling cascade.

Our enhanced sampling simulations found a stable conformation that is different to the native conformation, which we denoted as state B. Interestingly, this newly identified state B was PSK-dependent, i.e., sampled only when PSK was bound. A similar albeit unstable configuration was sampled for the apo configuration of PSKR1. This meta state found in the apo simulation may help to elucidate the folding timeline for the ID -loop: The apo PSKR1-ID loop folds into a conformation similar to state B, which can partially bind PSK. In sequence, the partially bound PSK-ID loop folds into the functional configuration.

The PSK directly affects the unfolding free energy landscape for the ID loop. The unfolding barrier between the native conformation and transition conformation that leads to state B has a value of 26 kcal/mol. The relative energy of state B in comparison to the native state is 22 kcal/mol. The application of a similar sampling method without PSK results in a reduction of the barrier height to 13 kcal/mol. Several metastable states were detected between the folded holo-structure and a higher radius of gyration, with no noticeable barrier. This energy landscape is characteristic of disordered regions, which shows several local energy minima and low transition energy barriers between them.

A novel “druggable” binding site was found in state B composed of residues located on the other face of the ID-domain, opposite to the PSK binding area. This pocket has an amphiphilic characteristic. Hence, this pocket may be used for the design of small molecular modulators for an antagonist effect on the PSK signalling cascade, working as a “molecular clamp” between the loop and the LRR domain. The fragments reported by FTMAP could be used as a starting point for a fragment-based ligand design campaign to design novel agrochemicals or molecular probes. Application of techniques commonly used in drug design, such as structure-guided fragment growth, scaffold hopping, and ultra-fast high-throughput virtual screening, could aid the design of potent and highly specific small molecule ligands. Our fragment docking indicated the specific regions to be explored by hydrophobic moieties in combination with hydrogen bonds formed between fragments and specific residues, i.e., Q517. In addition, this pocket may be used as a focal point in the studies of antagonistic effects of certain endogenous molecules on PSKR1.

Unexpectedly, we found IAA among the fragments interacting with the site identified at PSKR1. IAA is the most potent and abundant auxin natively occurring and functioning in plants. We found that other auxins were also binding to this site at the PSKR1. Even though the links between auxin and PSK systems have been published [22,23], direct interactions between auxins and PSK receptors have not been showed to date. Our results raise the question whether PSKR1 could be allosterically modulated by the high concentration of auxin, and need to be followed up by experimental works. Our results pave the way to validate a very exciting hypothesis, contributing to understanding the cross-talks between two important hormonal signalling systems in plants.

As previous studies have shown, PSK signalling is crucial for plant growth in several different species. PSK is one of the key hormones on cell elongation [9], working alongside the brassinosteroid signalling pathway. Furthermore, the PSK signalling cascade has been shown to be directly related to pollen tube growth [10]. Hence, controlling the PSKR1 function may lead to control of plant growth and reproduction. However, this may affect the immune system of the plant cell, since pathogen-associated molecular pattern (PAMP)-related genes are promoted by PSK [12,24].

Recently, Aghdam and co-workers showed that the use of exogenous PSK has a beneficial effect on fruit storage for both strawberries [25] and broccoli [26]. This opens a new window of possibilities regarding possible applications for PSK and the ID loop modulation on PSKR1. Therefore, we expect that by using molecules that could tune the ID loop between the native state and state B, we could tune the PSK signalling. This would allow for either a positive effect, i.e., prolonged cold storage of perishable crops, or an antagonistic effect on invasive crop growth.

## 4. Materials and Methods

### 4.1. Molecular Modelling

The holo *A. thaliana* PSKR1-PSK complex simulation started from the experimental structure (PDB code: 4Z63). The unstructured apo PSKR1 from *Daucus Carrota* started from the structure reported in the PDB code: 4Z62. Both models have been checked for any missing residues or double-occupancy atoms, which were amended. Any missing segments were modelled using MODELLER [27] interface in UCSF Chimera [28]. All glycosylated residues were kept.

### 4.2. Equilibrium Molecular Dynamics (MD) Simulations

All simulations were performed using GROMACS 2019 [28]. The equilibrium simulations were run in 3 sets: (1) An apo unstructured loop from *Daucus Carrota* (DcPSKR1), (2) a holo PSKR1-PSK from *Arabidopsis thaliana* (PSKR1), and (3) an apo configuration generated after removing the PSK from *A. thaliana holo*-PSKR1. All three systems were parametrised using the CHARMM36m [29] force field with automated identification of the glycans [30,31,32] and immersed in the cubic box of TIP3P [33] water model using CHARMM-GUI web server [34]. Box distance was set to 1 nm, and standard 3D periodic boundary conditions (PBC) were applied. The box was solvated, and Na^+^ and Cl^−^ ions were added to achieve a 0.1 M/L concentration and to maintain charge neutrality of the unit. The solvated systems were energy minimised and equilibrated. The minimisation ran using steepest descent for 1000 cycles followed by the conjugate gradient. Energy step size was set to 0.001 nm and the maximum number of steps was set to 50,000. The minimisation was stopped when the maximum force fell below 1000 kJ/mol/nm using the Verlet cutoff scheme. Treatment of long-range electrostatic interactions was set to Particle Mesh-Ewald (PME) [35], and the short-range electrostatic and van der Waals cut-off set to 1.0 nm. After the energy minimisation, heating to 300 K was performed for 20 ps with a time step of 2 fs and position restraints applied to the backbone in an NVT ensemble. The constraint algorithm used was LINCS [36], which was applied to all bonds and angles in the protein. With the Verlet cut-off scheme and the non-bonded short-range interaction, the cut-off was set to 1.0 nm. Long-range electrostatics were again set to PME. The temperature coupling was set between the protein and the non-protein entities by using a Berendsen thermostat, with a time constant of 0.1 ps and the temperature set to reach 300 K with the pressure coupling off. Pressure equilibration was run at 300 K with a Parrinello-Rahman pressure coupling on and set to 1 bar [37] in an NPT ensemble. The equilibration trajectories were set to three triplicates of 400 ns, with the first 10 ns discarded from the analysis.

Analysis of the trajectories was performed using GROMACS tools, including root-mean-square deviation (RMSD) to assess overall stability, per-residue root-mean-square fluctuation (RMSF) to assess the local flexibility, and solvent-accessible surface areas.

### 4.3. Island Domain Loop Well-Tempered Metadynamics

To evaluate the folding/unfolding of the island domain (ID) loop, well-temperated metadynamics [38] were performed using PLUMED [39,40,41,42]. Starting from the equilibrated crystal structure after the NPT equilibration, 5 × 10^7^ simulation steps were made. All MD parameters were the same as in the previous section. The well-tempered metadynamics used the radius of gyration (RG_ID_) for the atoms 7194–7760 (the ID loop) as the collective variable. The gaussian sigma was set to 0.1 nm, with a height of 1 kJ/mol with a deposition every 500 steps. The bias factor was set to 20, at a constant temperature of 300 K. A lower wall boundary was set for RG_ID_ at 1.27 nm (its initial value), with a height of 2000 kJ/mol. The convergence evaluation was made via time and Gaussian deposition convergence. The mapping of potential novel binding spots was performed using FTMap (URL: https://ftmap.bu.edu/, Accessed 14 October 2020) [43], a well-established web server for mapping of allosteric binding sites. To evaluate the binding of druglike fragments from Enamine Essential fragment library, which contains 320 druglike fragments used for hotspot characterization. The molecular docking to the state B binding site was made using SeeSAR [44]. 325 fragments were docked, using 50 different poses for each fragment, using high clash tolerance filter. The binding affinity and torsional suitability were evaluated using HYDE algorithm [45].

## 5. Conclusions

In this work, we evaluated the stability effect of PSK on the PSK receptor 1 through molecular dynamics simulations and enhanced sampling techniques. By using equilibrium dynamics and well-tempered metadynamics, we showed a novel and stable configuration on PSKR1, which is “druggable” by small molecule modulators. We proposed a model of how PSK binds to PSKR1 via sequential folding states of the ID loop. We expect that this model would help to clarify PSK-PSKR1-SERK1 complex formation and show how modulating the ID loop dynamics may regulate the PSK signalling pathway.

## Figures and Tables

**Figure 1 ijms-22-01806-f001:**
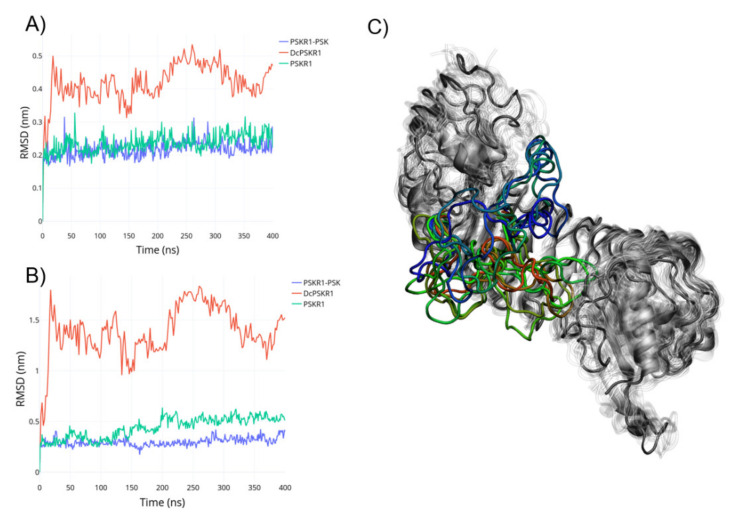
(**A**) Calculated root-mean-square deviation (RMSD) for the whole structure with the leucine-rich repeat (LRR) domain fitted. (**B**) Calculated RMSD for the ID loop with the LRR domain fitted. (**C**) Ensemble from the DcPSKR1 dynamics; grey: The LRR domain, and transparent grey: The overlayed configurations; the rainbow gradient: The disordered conformations of the ID loop from DcPSKR1.

**Figure 2 ijms-22-01806-f002:**
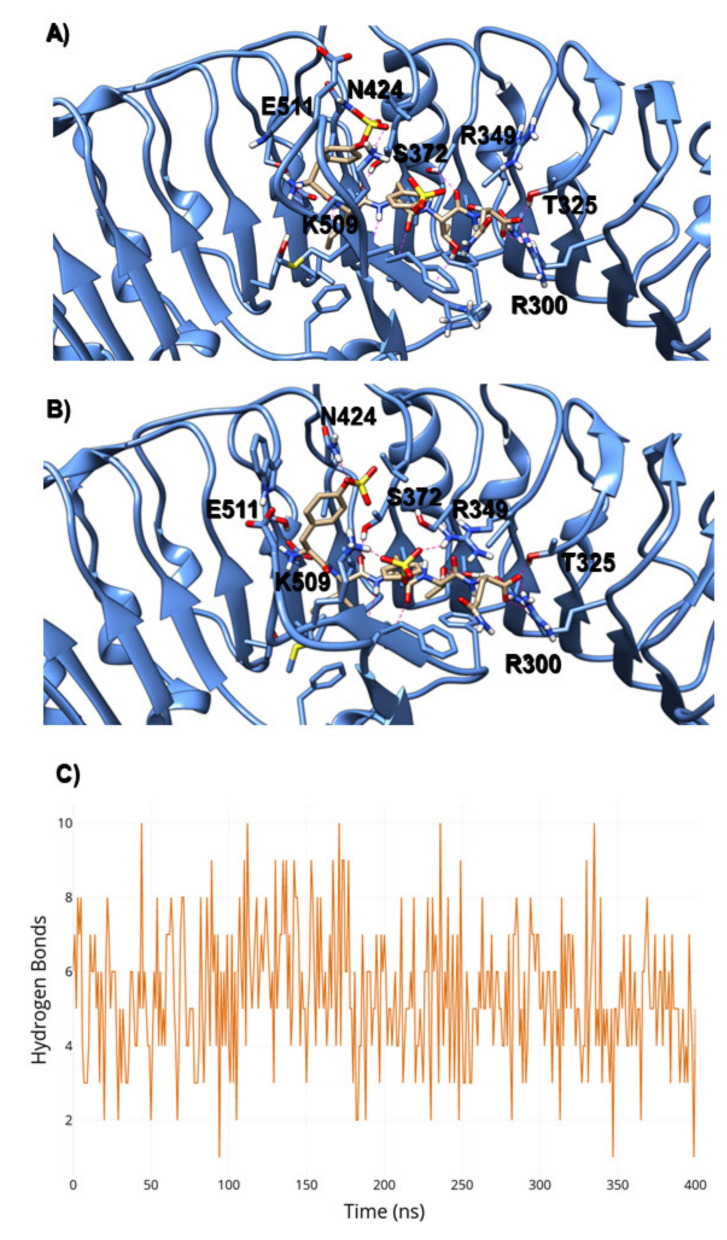
(**A**) Residue interactions between PSKR1-PSK from crystal structure. (**B**) Residue interactions between PSKR1-PSK after 400 ns of molecular dynamics (MD) simulation. (**C**) Hydrogen bonds between PSK and PSKR1 through the simulation time.

**Figure 3 ijms-22-01806-f003:**
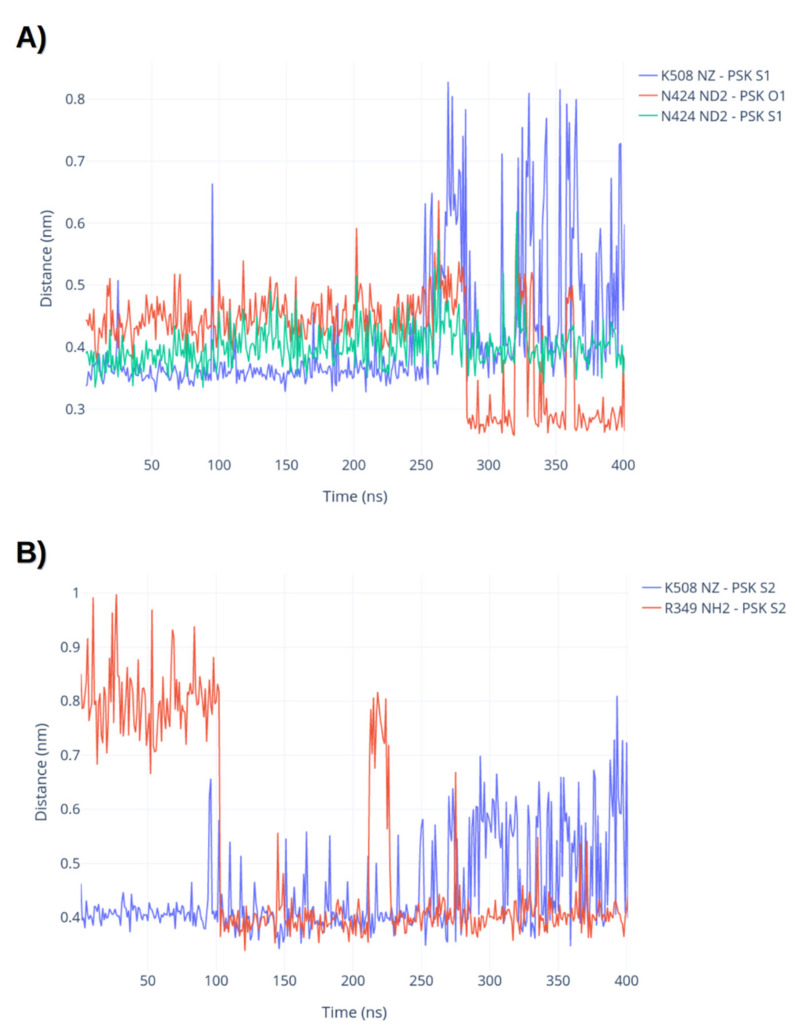
(**A**) Distances between atoms on the N-terminal sulphated tyrosine and PSKR1 during the simulation time. (**B**) Distances between atoms on the C-terminal sulphated tyrosine and PSKR1 during the simulation time.

**Figure 4 ijms-22-01806-f004:**
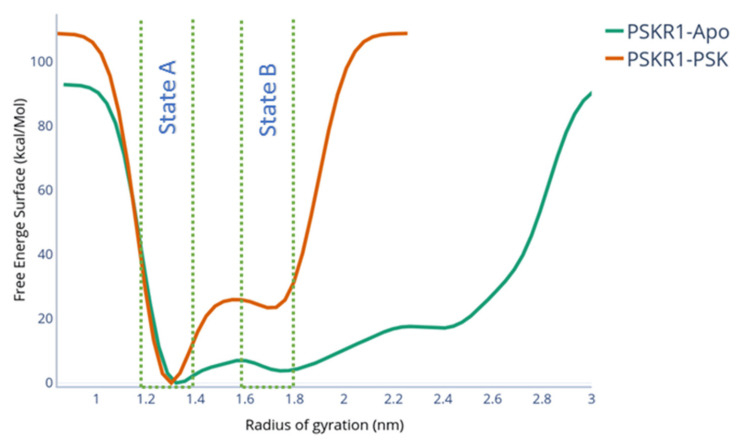
Free energy surface calculated by metadynamics for PSKR1-PSK and PSKR1-apo.

**Figure 5 ijms-22-01806-f005:**
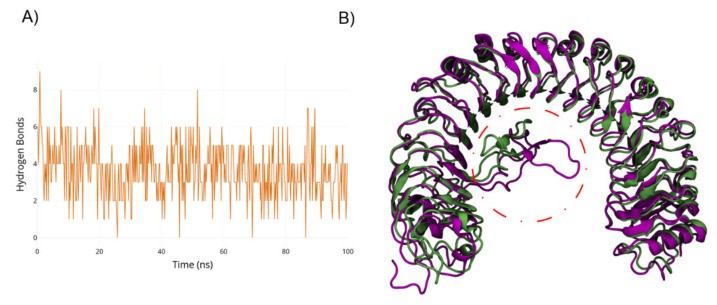
(**A**) Hydrogen bonds between PSKR1 and PSK during the 100 ns of metadynamics. (**B**) The native conformation of the PSRK1-PSK is coloured green; the state B conformation is coloured purple. The ID loop is circled by the red dashed line.

**Figure 6 ijms-22-01806-f006:**
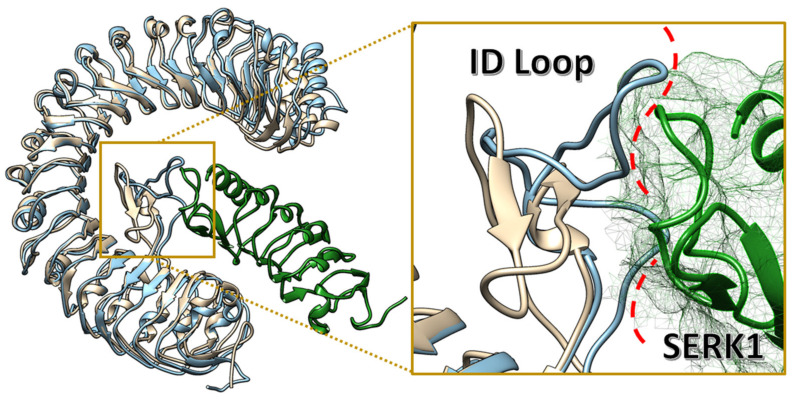
State B represented in blue, in ochre the PSRK1 crystal structures, and in green connected spheres. In focus: The red dashed lines represent regions with molecular clashes between the ID loop and SERK1.

**Figure 7 ijms-22-01806-f007:**
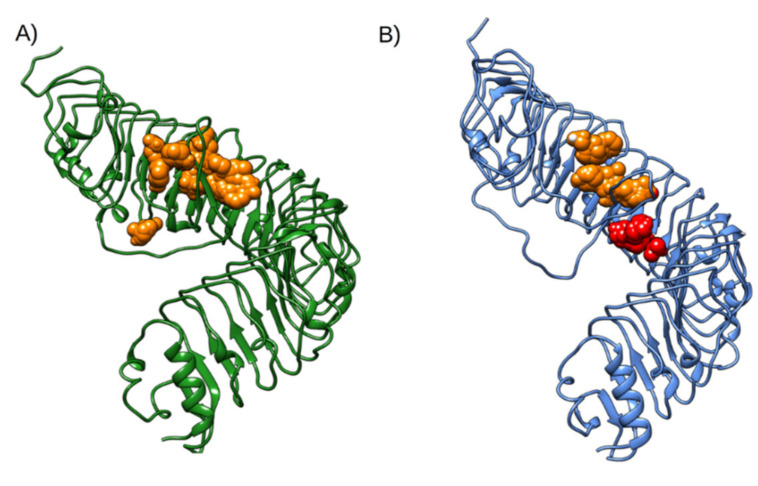
(**A**) The PSKR1 average structure obtained during 400 ns simulation; FTMAP cluster indicating the PSK binding site is coloured orange. (**B**) PSKR1 in state B; FTMAP cluster indicating the PSK binding site is coloured orange; the novel allosteric binding site is coloured red.

**Figure 8 ijms-22-01806-f008:**
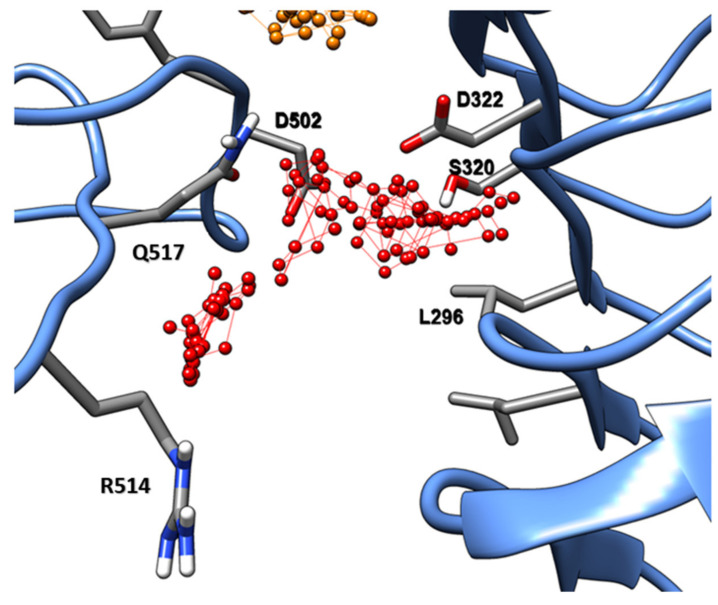
State B binding site represented by the red connected spheres. In focus: The residues involved in the binding pocket formation.

**Figure 9 ijms-22-01806-f009:**
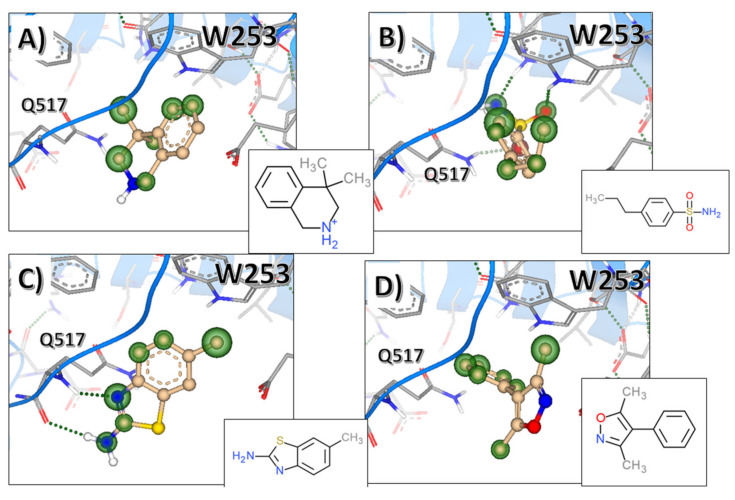
Molecular docking results for the top four fragment-like molecules with their respective 2D structures: (**A**) 4,4-dimethyl-1,2,3,4tetrahydroisoquinoline, (**B**) 4-propylbenzenesulfonamide, (**C**) 2-amino-6-methylbenzothiazole, (**D**) 3,5-dimethyl-4-phenyl-isoxazole. 2D chemical structures are showed in insets. Green halos indicate favourable interactions. Hydrogen bonds are indicated by black dashed lines.

## Data Availability

Data available upon request.

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
