# Peer review of "Controlling the Heterodimerisation of the Phytosulfokine Receptor 1 (PSKR1) via Island Loop Modulation"

_ijms, 2021, doi:10.3390/ijms22041806_

Round 1

Reviewer 1 Report

Review of the article: "Controlling the heterodimerisation of the phytosulfokine receptor 1 (PSKR1) via island loop modulation".

In my opinion, this is an interesting and potentially valuable article. The weakness of the article is that it does not present specific applications of the obtained results, especially, in what way the information about a novel "druggable" binding site can be used to improve plant development and growth.

Moreover, the article does not present utilization of the model, instead, the article appears unclear information, for example:
a) "We expect that this model would help to clarify PSK-PSKR1-SERK1 complex formation and show how modulating the ID loop dynamics may regulate the PSK signalling pathway" (in the Conclusions) - based on the model, this clarification of PSK-PSKR1-SERK1 complex formation and presentation how modulating the ID loop dynamics may regulate the PSK signalling pathway should be presented in the article.
b) "We expect that our results will open new ways..." (in the Abstract) - examples of these new ways should be presented.

Author Response

On behalf of my coworkers and myself, I would like to thank the Reviewer 1 for assessing our manuscript and for providing some very helpful comments. All the corrections suggested were applied through the revised manuscript. Responses and clarifications to further specific comments are given below.

"In my opinion, this is an interesting and potentially valuable article. The weakness of the article is that it does not present specific applications of the obtained results, especially, in what way the information about a novel "druggable" binding site can be used to improve plant development and growth. Moreover, the article does not present utilization of the model, instead, the article appears unclear information, for example:
a) "We expect that this model would help to clarify PSK-PSKR1-SERK1 complex formation and show how modulating the ID loop dynamics may regulate the PSK signalling pathway" (in the Conclusions) - based on the model, this clarification of PSK-PSKR1-SERK1 complex formation and presentation how modulating the ID loop dynamics may regulate the PSK signalling pathway should be presented in the article.
b) "We expect that our results will open new ways..." (in the Abstract) - examples of these new ways should be presented."

To clarify this, Figure 6 was added, which shows the overlap of state B with the crystallographic structure of PSKR1-PSK-SERK1 heterocomplex. This figure shows that in state B, ID loop overlaps with the position of SERK1, creating significant steric clashes, which destabilise the complex.

To provide the proof-of-concept data for the model applications, we added our pilot molecular docking data on 325 synthetic small molecules, including auxins.  The works, which targeted the State B binding site, shows a direction to structure-guided ligand design using this novel binding site. We also demonstrated that the current work may be applicable in data-driven hypothesis testing, i.e. to assess whether PSK receptors may be allosterically modulated by auxins and to validate a putative binding site, via e.g. site-directed mutagenesis experiments.

These results were added to the manuscript, with the small molecules amenable for experimental testing and scaffold expansion are listed in Figure 9. The binding mode of indole-3-acetic acid (IAA) is showed in the new Supplementary Figure 6. In addition, we clarified/rectified areas if interest exemplified by the reviewer comments.

I hope that the response thoroughly addresses all concerns regarding the results and their discussion, data analysis and presenting of the data, and I am looking forward to the feedback.

Yours Sincerely,

Dr. Joao Victor de Souza

Reviewer 2 Report

In this MS, the authors aimed to decipher the binding event of PSK with PSKR1. For this, they have used rigorous theoretical tools such as molecular dynamics (MD) simulations and free energy calculations. The method used is well established and the experiments are well planned. My minor comments are-

  1. In my opinion, figure 1 is not required to be shown in the introduction section. The authors are already supporting this with the previously published studies, hence ideally they should delete it from the introduction part.
  2. The limitation of MD is the validity of the result. Assessing ergodicity from the trajectory and determining whether convergence has been achieved is the subject of concern. The author must give some statement on the limitation to their finding.

Author Response

On behalf of my coworkers and myself, I would like to thank the Reviewer 2 for assessing our manuscript and for providing some very constructive comments. All the corrections suggested were applied through the revised manuscript. Responses and clarifications to further specific comments are given below.

In this MS, the authors aimed to decipher the binding event of PSK with PSKR1. For this, they have used rigorous theoretical tools such as molecular dynamics (MD) simulations and free energy calculations. The method used is well established and the experiments are well planned. My minor comments are-

1 - In my opinion, figure 1 is not required to be shown in the introduction section. The authors are already supporting this with the previously published studies, hence ideally they should delete it from the introduction part.

We appreciate the comment, and we deleted theFigure 1 following the reviewer suggestion.

2 - The limitation of MD is the validity of the result. Assessing ergodicity from the trajectory and determining whether convergence has been achieved is the subject

This has been clarified and explained in the methods part of the paper, with the convergence analysis of the metadynamics (time convergence and bias convergence, respectively), and the convergence of the equilibrium runs (RMSD vs time for all runs and replicas) shown in the supplementary data (Figure S5)

I hope that the response thoroughly addresses all concerns regarding the results and their discussion, data analysis and presenting of the data, and I am looking forward to the feedback.

Yours Sincerely,

Dr. Joao Victor de Souza

Round 2

Reviewer 1 Report

Thank you for these corrections.
The Authors addressed correctly to all my comments and concerns. Now the article is much better and in my opinion it can be published in International Journal of Molecular Sciences.